# Decoding Functional High-Density Lipoprotein Particle Surfaceome Interactions

**DOI:** 10.3390/ijms23169506

**Published:** 2022-08-22

**Authors:** Kathrin Frey, Sandra Goetze, Lucia Rohrer, Arnold von Eckardstein, Bernd Wollscheid

**Affiliations:** 1Institute of Translational Medicine, Department of Health Sciences and Technology, ETH Zurich, 8093 Zurich, Switzerland; 2Institute for Clinical Chemistry, University Hospital Zurich, 8091 Zurich, Switzerland; 3Swiss Institute of Bioinformatics (SIB), 1015 Lausanne, Switzerland; 4ETH PHRT Swiss Multi-Omics Center (SMOC), 8093 Zurich, Switzerland

**Keywords:** HDL, molecular health, signaling, surfaceome, receptome, ligand–receptor interactions, spatial proteotyping, chemoproteomics

## Abstract

High-density lipoprotein (HDL) is a mixture of complex particles mediating reverse cholesterol transport (RCT) and several cytoprotective activities. Despite its relevance for human health, many aspects of HDL-mediated lipid trafficking and cellular signaling remain elusive at the molecular level. During HDL’s journey throughout the body, its functions are mediated through interactions with cell surface receptors on different cell types. To characterize and better understand the functional interplay between HDL particles and tissue, we analyzed the surfaceome-residing receptor neighborhoods with which HDL potentially interacts. We applied a combination of chemoproteomic technologies including automated cell surface capturing (auto-CSC) and HATRIC-based ligand–receptor capturing (HATRIC-LRC) on four different cellular model systems mimicking tissues relevant for RCT. The surfaceome analysis of EA.hy926, HEPG2, foam cells, and human aortic endothelial cells (HAECs) revealed the main currently known HDL receptor scavenger receptor B1 (SCRB1), as well as 155 shared cell surface receptors representing potential HDL interaction candidates. Since vascular endothelial growth factor A (VEGF-A) was recently found as a regulatory factor of transendothelial transport of HDL, we next analyzed the VEGF-modulated surfaceome of HAEC using the auto-CSC technology. VEGF-A treatment led to the remodeling of the surfaceome of HAEC cells, including the previously reported higher surfaceome abundance of SCRB1. In total, 165 additional receptors were found on HAEC upon VEGF-A treatment representing SCRB1 co-regulated receptors potentially involved in HDL function. Using the HATRIC-LRC technology on human endothelial cells, we specifically aimed for the identification of other bona fide (co-)receptors of HDL beyond SCRB1. HATRIC-LRC enabled, next to SCRB1, the identification of the receptor tyrosine-protein kinase Mer (MERTK). Through RNA interference, we revealed its contribution to endothelial HDL binding and uptake. Furthermore, subsequent proximity ligation assays (PLAs) demonstrated the spatial vicinity of MERTK and SCRB1 on the endothelial cell surface. The data shown provide direct evidence for a complex and dynamic HDL receptome and that receptor nanoscale organization may influence binding and uptake of HDL.

## 1. Introduction

High-density lipoprotein (HDL) is the term describing a complex mixture of particles of different sizes, shapes, densities, and compositions. HDL particles contain about 300 proteins [1], a similarly large number of lipid species, and non-coding RNAs [2]. Low plasma levels of HDL cholesterol are associated with an increased risk of mortality and other morbidities including atherosclerotic cardiovascular diseases, diabetes, chronic kidney disease, infections, and autoimmune diseases [3]. The causal role of HDL in the pathogenesis of these diseases is controversial in part due to the particle’s structural and functional complexity [4]. The classical function of HDL is the delivery of excess cholesterol from peripheral tissues, via lipid-laden macrophages (foam cells), to the liver for biliary excretion. This reverse cholesterol transport (RCT) involves interactions between HDL and cells of various types, for example, endothelial cells to travel between intravascular and extravascular compartments [5], lipid-laden macrophages (foam cells) for the induction of cholesterol efflux, and lastly hepatocytes for either selective uptake of cholesterol or holoparticle uptake before excretion [6]. Additionally, HDL is involved in many other signaling events unrelated to RCT such as the regulation of the endothelial barrier integrity, angiogenesis, vasoreactivity, and inflammation [5].

The molecular mechanisms involved in the binding of HDL to cells that result in particle or lipid uptake and/or signaling are poorly understood, partially because the inventory of HDL receptors appears to be still incomplete. SCRB1 is currently the only confirmed HDL receptor [7]. The binding of HDL to SCRBI mediates the selective uptake of lipids into hepatocytes as well as steroidogenic cells and facilitates cholesterol efflux from macrophages [7,8]. In endothelial cells, SCRB1 limits holoparticle uptake and mediates several signaling functions of HDL such as the stimulation of nitric oxide production, proliferation, migration, and progenitor cell differentiation, the inhibition of apoptosis, as well as the suppression of adhesion molecule expression and, hence, leukocyte diapedesis [5]. Given the multiplicity of signaling events that appear to be mediated by HDL, it is conceivable that HDL’s functionality is mediated by multiple receptor and intracellular adapter proteins [9]. It was recently demonstrated that S1PR1 transiently interacts with SCRB1 to trigger calcium flux and S1PR1 internalization [10]. Calcium flux through S1PR1 is triggered by the binding of the ligand S1P, which is enriched on HDL particles carrying APOM [11]. The binding of HDL’s core protein component APOA1 to ABCA1 and ecto-F1-ATPase (ATPK) elicits cholesterol efflux and the generation of ADP. The latter activates purinergic receptors to trigger HDL holoparticle uptake into hepatocytes and endothelial cells by an as-yet-unknown pathway [12,13]. Finally, platelet glycoprotein CD36, which shares high sequence similarity with SCRB1, reportedly binds HDL specifically on hepatocytes [14]. However, with the exception of SCRB1, which mediates selective lipid uptake, the interactions of HDL with these receptor proteins do not lead to specific HDL-mediated functionalities, suggesting the involvement of other co-receptors and receptor synapses. Therefore, and because of the complexity of HDL, we hypothesize that a complex and dynamic HDL receptome mediates HDL signaling.

To better understand the dynamic interplay between surfaceome-residing receptor neighborhoods and HDL functionality, we set out to characterize the HDL receptome using a combination of chemoproteomic technologies including automated cell surface capturing (auto-CSC) [15] and HATRIC-based ligand–receptor capturing (LRC) [16]. Both proteotyping technologies enable the mass-spectrometric-based identification and quantitation of N-linked glycosylated receptors at the cellular surface. While the auto-CSC technology enables the identification of the acute *N*-glycosylated surfaceome, the HATRIC-LRC technology enables the identification of an unknown receptor for a known ligand such as HDL via a trifunctional cross-linker-based strategy. To establish a cell surface atlas of potential HDL-interacting proteins, we applied auto-CSC to model systems mimicking tissues relevant for RCT and frequently used in HDL research. Tissue-specific receptor neighborhoods are considered hubs that translate information from the extracellular environment to the cell interior [17]. Such signaling hubs are highly dynamic [18] and are affected by external stimuli that in turn can influence ligand–receptor interactions [17]. We investigated the dynamic behavior of such receptor neighborhoods by treating human endothelial cells with VEGF-A, a known mediator of SCRB1 translocation to the cell surface [19], which led to major changes in the potential HDL-receptor interaction landscape. Finally, we sought to identify endothelial HDL receptors using HATRIC-LRC, which enables direct ligand–receptor capture via a trifunctional linker. The HATRIC-LRC experiments led to the discovery of tyrosine-protein kinase MERTK, a TAM receptor family member involved in the maintenance of vascular cell homeostasis [20], as a modulator of HDL binding and uptake.

## 2. Results

### 2.1. Characterization of the Potential Receptor Interaction Space of HDL

To better understand HDL–cell interactions, the underlying surfaceome must be defined. Employing auto-CSC [15], we set out to qualitatively and quantitatively characterize the cellular surfaceomes of model systems mimicking tissues relevant for RCT and frequently used in HDL research (Figure 1A,B): (i) the human endothelial somatic hybrid EA.hy926 cells, (ii) the primary human aortic endothelial cells (HAECs), (iii) the human hepatocyte cell line HEPG2, and (iv-vi) the human monocyte THP1 cells before and after activation with phorbol 12-myristate 13-acetate (PMA) and after their differentiation into foam cells upon treatment with acetylated LDL (acLDL) (Appendix A).

Based on the scaled rank of all quantified glycosylated proteins over all samples, the endothelial cells EA.hy926 cells and HAECs clustered in the principal component analysis (Figure 2A, Appendix A). Although closely related, we observed surfaceome remodeling during the differentiation of THP1 monocytes into THP1 macrophages and upon transformation into lipid-laden foam cells (Appendix A). As previously described, quantitative surface receptor differences induced by the stimulation of THP1 with PMA include the increased abundance levels of the activin receptor AVR2A, monocyte differentiation antigen CD14, the integrin ITAM, and the receptor tyrosine kinase MERTK, as well as the decreased abundance of T-cell surface glycoprotein CD4, the growth factor receptor KIT, and the mannose receptor MRC2 (Appendix A) [21]. In addition, the abundance of ABCA1 on the cellular surface increased during the differentiation of THP1 cells induced by treatment with acLDL, whereas the abundance of SCRB1 decreased upon PMA stimulation but increased again upon differentiation into foam cells. The oxidized LDL (oxLDL) receptor CD36 and the macrophage scavenger receptors MSRE1 and MSRE2 were slightly enriched on foam cells compared with THP1 monocytes and THP1 macrophages.

We identified 419, 496, 580, and 497 surfaceome proteins on EA.hy926, HEPG2, foam cells, and HAECs, respectively (Figure 2B, Appendix A). Of these, 155 proteins were detected on the surfaces of all four of these cell types, and 40 were identified exclusively on HAECs and EA.hy926 cells (Figure 2B, purple bar). One of the proteins identified only on endothelial cells was S1PR1, which is known to mediate several effects of HDL on endothelial cells including barrier integrity, nitric oxide production, and suppression of leukocyte adhesion [22]. The main HDL receptor SCRB1 was identified on all four cell types and most abundantly on HEPG2 cells (Figure 2C). Other receptors involved in cholesterol- or lipoprotein-related biological processes were also differentially abundant on the different surfaceomes. ABCA1 and CD36, for instance, were predominantly detected on foam cells and HEPG2 cells but were less abundant or were not detected on the endothelial cell lines. Furthermore, we detected apolipoprotein APOB on both hepatocytes and foam cells and apolipoprotein APOM and glycoprotein APOH on hepatocytes. Hepatocytes are known to synthesize and secrete apolipoproteins [23], whereas APOB on foam cells is most likely derived from the acLDL treatment. PLTP, which is known to mediate the transfer of phospholipids and free cholesterol to HDL [24], was identified on all four RCT-relevant cellular model systems but was only slightly above the lower limit of quantification on HEPG2 cells and HAECs. The different characteristics of the surfaceome landscapes of these models suggest the tissue-specific encoding of HDL functionality through receptor neighborhoods.

### 2.2. VEGF-A Treatment Triggers Reorganization of the Surfaceome of HAECs

The cellular surfaceome is not a static organization of receptors and lipids. It is continually exposed to extracellular stimuli and, therefore, reacts and adapts to environmental changes. VEGF-A triggers the translocation of SCRB1 from an intracellular pool to the plasma membrane and, as a consequence, HDL uptake by HAECs [19]. As VEGF-A might also affect the availability of additional co-receptors of HDL on the cellular surface, we assessed surfaceome changes on HAECs treated with VEGF-A. Although SCRB1 was the most prominently affected protein, we observed a significant quantitative reorganization of 165 additional cell surface receptors (Figure 3A, Appendix A). Like SCRB1, MERTK was upregulated in the VEGF-A-treated condition. In contrast, PLTP, S1PR1, and S1PR2 were downregulated. Unexpectedly, the decoration of the cellular surface with the VEGF-A receptor VGFR2 was not affected by the presence or absence of its ligand.

To globally assess the functional processes of up- or downregulated protein groups, we performed a Gene Ontology (GO) enrichment analysis of biological processes and molecular function (Appendix A). This analysis revealed 23 significantly enriched GO terms for the group of upregulated proteins and 12 significantly enriched terms in the group of downregulated proteins. The most enriched GO terms, according to the family-wise error rate, were G protein-coupled receptor binding, which was enriched in the group of downregulated proteins, and virus receptor activity, which was enriched in the group of upregulated proteins (Figure 3B). The proteins associated with the highest numbers of terms were the integrin ITAV (15 terms), SCRB1 (13 terms), and EGFR (13 terms). About 80% of all proteins were specifically associated with three or fewer GO terms. This analysis showed that VEGF-A not only influences the surface abundance of SCRB1 on HAECs but also quantitatively modulates a large fraction of the cellular surfaceome landscape, which presumably has functional implications.

### 2.3. Endothelial MERTK Is a co-Receptor of HDL, Resides Proximal to SCRB1, and Contributes to HDL Binding and Uptake

To identify novel direct interactors of HDL on the cellular surface, we performed a HATRIC-LRC experiment on EA.hy926 cells as previously described [16]. EA.hy926 cells were incubated with either lipid-free APOA1, a minimal artificial HDL particle (rHDL) reconstituted from APOA1 and palmitoylphopshatidylcholine (POPC) in a 1:80 molar ratio, or native HDL as ligands. Although lipid-free APOA1 still binds to SCRB1 on human endothelial cells, it does so with a much lower affinity than does APOA1 complexed with lipids [25]. Both rHDL and native HDL are fully functional and strongly bind to SCRB1. The native HDL is more complex than rHDL (Appendix A and [1]) and presumably mediates more ligand–receptor interactions at the cellular surface. As a negative control, we included TRFE in our HATRIC-LRC experiment; TRFE is the ligand for the transferrin receptor TFR1, a receptor that has not been shown to be involved in HDL signaling. As expected, in our TRFE control, TRFE and its receptor TFR1 were highly enriched in comparison to APOA1, rHDL, and native HDL conditions (Figure 4A, Appendix A). In cells treated with APOA1, rHDL, and native HDL conditions, we identified HDL proteins that were significantly enriched compared with the TRFE control (Figure 4A, Appendix A). Enrichment increased as the ligand complexity increased. These HDL-derived proteins were most likely enriched due to the labeling of the ligand (i.e., HDL) itself. With lipid-free APOA1 as a ligand, we also identified APOC3, most likely an artifact of the APOA1 purification. We did not identify ABCA1 under any condition, most likely because of its relatively low abundance on the cellular surface of EA.hy926 cells (Figure 2C). Eleven proteins were identified on cells treated with reconstituted minimal HDL and with native HDL as ligands. Nine are annotated as cell surface proteins including MERTK and the scavenger receptors SCRB1 and SCAR3.

In the rHDL HATRIC-LRC condition, we identified the phospholipid-transporting ATPase ABCA7 and the protocadherin PCDC1 as potential HDL co-receptors. ABCA7 was previously reported to bind APOA1 to mediate phospholipid efflux from cells [26]. In the LRC-HATRIC comparison of native HDL with TRFE as ligand, we identified two surface proteins that are potentially part of the HDL receptome, the sodium/calcium exchanger NAC2 and multiple epidermal growth factor-like domains protein 8 (MEGF8). Another MEGF family member and scavenger receptor, MEGF10, was shown to bind the complement protein C1Q, which, in turn, was shown to associate with HDL [27,28]. Whether MEGF8 also acts as a scavenger receptor remains to be determined.

In addition to the significant enrichment observed in HATRIC-LRC experiments with both rHDL and native HDL (Figure 4A, Appendix A), MERTK was also upregulated, together with SCRB1, upon VEGF-A treatment of HAECs (Figure 3A, Appendix A). MERTK is a receptor tyrosine kinase that regulates many physiological processes including cell survival, migration, differentiation, and phagocytosis of apoptotic cells [20]. It was also shown to mediate efferocytosis in atherosclerotic lesions [29].

Therefore, we selected MERTK as a potential HDL receptor candidate for follow-up validation experiments to confirm its role as an HDL (co-)receptor. Using proximity ligation assay (PLA), we confirmed that SCRB1 and MERTK localize in the same neighborhood on the surface of both EA.hy926 cells and HAECs (Figure 4B). To show the functional relevance of MERTK in the context of HDL binding and uptake, we suppressed *MERTK* expression using agents that mediate RNA interference (Appendix A). HDL binding and uptake were significantly reduced in both endothelial cell types after MERTK depletion to an extent similar to that observed after *SCRB1* silencing (Figure 4C,D). Ligand binding to the MERTK receptor induces the autophosphorylation of the protein on its intracellular domain, providing a docking site for downstream signaling molecules [30]. However, when we measured the level of MERTK phosphorylation before and after HDL binding in endothelial cells, we did not detect a difference in the phosphorylation state of the receptor (Appendix A). Thus, our data indicate that both SCRB1 and MERTK can modulate HDL binding and uptake and that the phosphorylation of MERTK is not likely necessary for this activity.

## 3. Discussion

HDL, a multimolecular complex of proteins and lipids, exerts a broad spectrum of functions in different cell types. It is conceivable that the molecular mode of action of HDL with cell surface proteins is many-to-many rather than one-to-one or many-to-one. We, therefore, set out to characterize the cellular interaction space of HDL using chemoproteomic technologies. Our cell surface protein atlas provides the most comprehensive resource of the dynamic and HDL-relevant surfaceome to date, which can serve as a base for the deconvolution of tissue-specific HDL signaling mechanisms. Additionally, it provides the first step toward an understanding of what we demonstrate are many-to-many interactions. We identified around 500 cell surface proteins in each of the 4 cellular models of RCT: EA.hy926 cells, HAECs, HEPG2 cells, and foam cells resulting from differentiation of THP1 cells. Of these, only 155 surface proteins were shared by all 4 cell types. The cellular surfaceome differences imply different functionalities, in line with the known tissue-specific roles of HDL.

Of the total 1054 proteins characterized across the 4 RCT-relevant cell types, more than 30% were not annotated in Surfy, an in silico surfaceome resource [31], which highlights the importance of our experimental cell surface protein atlas. Using auto-CSC, we also captured proteins that are not linked directly to the plasma membrane via transmembrane domains or glycosylphosphatidylinositol anchors but that were associated through interactions with other components of the plasma membrane. Some of these proteins are of high relevance for HDL formation and remodeling. For instance, we identified PLTP on all four cell types. HDL-particle-bound PLTP transfers phospholipids from triglyceride-rich particles to HDL and remodels lipid-poor and protein-rich HDL3 into lipid-rich and protein-poor HDL2 [32]. Cell-bound PLTP within atherosclerotic plaques may serve as a bridging protein to mediate the association of HDL with the extracellular matrix [33]. The auto-CSC technology only captures cell surface proteins carrying extracellular *N*-glycosylation motifs, which comprise the majority of the cellular surfaceome: According to predictions based on Surfy, fewer than 5% of cell surface proteins do not contain an extracellular *N*-glycosylation motif [31]. Nevertheless, non-*N*-glycosylated proteins such as ATPK or ABCG1 also contribute to the uptake of HDL by hepatic and endothelial cells [12,13,25]. The analysis of such proteins with respect to HDL functionality will require different technological approaches.

Extrinsic factors can trigger changes in the cellular surfaceome and, thereby, HDL-related functionalities. For example, we demonstrated that there is extensive, quantitative remodeling of the endothelial cell surface proteome upon VEGF-A treatment. Corroborating previous findings of our lab [19], SCRB1 was one of the most VEGF-A-responsive proteins [19]. Interestingly, the majority of receptors upregulated together with SCRB1 upon VEGF-A treatment are associated with the Gene Ontology traits of host entry and virus receptors. VEGF-A treatment promotes vaccinia host entry via the activation of the AKT pathway [34], the same mechanism that was also shown to be of relevance for HDL cell endocytosis [19]. HDL particles share structural similarities with lipid-coated viruses in terms of size, mixed protein and lipid cell surface composition, and the cargo of RNA. During infection, viruses and HDL might both co-opt endocytic mechanisms and pathways. Notably, the scavenger receptor family is known to be targeted by different viral particles [27]. In particular, SCRB1, the main HDL receptor, is also involved in hepatitis C virus entry into cells [35], and, as recently shown, SARS-CoV-2 entry is HDL-dependent [36].

The VEGF-A treatment of HAECs decreased the abundance levels of several cell surface proteins, including the sphingosine-1-phosphate receptors S1PR1 and S1PR2. This finding contrasts with that of a previous report indicating that VEGF-A induces the mRNA expression of *S1PR1* [37]. We found that S1PR3 was enriched in the VEGF-A-treated cells. Both S1PR1 and S1PR3 were previously shown to mediate the effects of HDL on endothelial barrier integrity, nitric oxide production, and leukocyte diapedesis [22]. As VEGF-A stimulation promotes angiogenesis, cell proliferation, and migration [38,39], we assume that the ligands S1P and VEGF-A balance each other, to secure endothelial integrity and functionality.

Using HATRIC-LRC, we set out to identify the cell surface proteins that interact with HDL on human endothelial cells. We identified MERTK as a novel co-receptor critical for HDL binding and uptake. MERTK was present on the surfaces of all investigated cell types. In macrophages, MERTK was reported to influence atherosclerosis progression through apoptotic cell clearance [40]. In THP1 cells, MERTK mitigates MSRE abundance upon its interaction with protein S, resulting in decreased acLDL uptake [41]. Both MERTK and HDL mitigate the inflammatory response triggered by lipid-laden macrophages [42]. These findings support a MERTK-dependent link between lipid metabolism and inflammation [20]. Endothelial MERTK has been reported to contribute to the maintenance of endothelial barrier function in human lung microvascular endothelial cells [43]. Furthermore, like HDL, MERTK inhibits neutrophil trans-endothelial migration in vitro [43]. Finally, MERTK facilitates the cellular entry of filoviruses [44], again supporting the hypothesis that viruses and HDL share cellular entry routes.

Ligand binding to the MERTK receptor induces its autophosphorylation, but we did not detect a difference in the phosphorylation state of the receptor in the presence of HDL. This might be an indication that HDL-mediated functionalities do not involve intracellular MERTK phosphorylation. MERTK may modulate HDL binding and uptake through extracellular interactions with HDL and co-receptors such as SCRB1.

HATRIC-LRC allows the identification of direct ligand–receptor interactions, but this technique does not provide any information about a receptor’s functional neighborhood. Proximity labeling technologies such as LUX-MS [45] have the potential to decipher such nanoscale organizations. Using proximity ligation experiments, we showed that MERTK and SCRB1 are close in space on the epithelial cell membrane. Delineating the molecular organization of the cellular surfaceome and its interactions with HDL is a prerequisite for understanding the communication of HDL with cells, including lipid fluxes, signaling, and internalization. This may help to pave the way for exploiting HDL as a target for new treatment and prevention strategies for a wide range of pathologies, including cardiovascular disease. The complexity of HDL suggests that different HDL sub-pools exist that target distinct receptors and neighborhoods with specific molecular functionality. This might also explain the wide range of functionalities exerted by HDL. Our comprehensive analysis of the dynamic and HDL-relevant surfaceome will serve as a base for the deconvolution of tissue-specific HDL signaling mechanisms.

## 4. Materials and Methods

### 4.1. Cell Culture

HAECs (304-05a, Cell Applications Inc., San Diego, CA, USA) were grown in a Human Endothelial Cell Growth Medium, all-in-one, ready-to-use (Clonetics CC-3156, LONZA, Basel, Switzerland). For auto-CSC experiments with and without VEGF-A, HAECs were cultured in EBM-2 supplemented with SingleQuots (Clonetics CC-4176, LONZA, Basel, Switzerland) and an Endothelial Cell Growth Kit-VEGF (PCS-100-041, containing hFGF, hVEGF-A, hIGF-1, hEGF, hydrocortisone, ascorbic acid, and heparin, ATCC, Manassas, VA, USA) with or without VEGF-A for 72 h. EA.hy926 cells (CRL-2922, ATCC, Manassas, VA, USA) and HEPG2 cells (HB-8065, ATCC, Manassas, VA, USA) were grown in Dulbecco’s modified Eagle’s medium DMEM (Sigma-Aldrich, St. Louis, MO, USA) containing 10% fetal bovine serum (FBS, Sigma-Aldrich, Buchs, Switzerland) and 1% penicillin–streptomycin (Sigma-Aldrich, St. Louis, MO, USA). THP1 cells (TIB-202, ATCC, Manassas, VA, USA) were grown in an RPMI 1640 medium ( Sigma-Aldrich, St. Louis, MO, USA) containing 10% FBS and 1% penicillin–streptomycin. THP1 cells were activated with 50 ng/mL PMA. After 48 h, the medium was changed to an RPMI 1640 medium containing 10% FBS and 1% penicillin–streptomycin for a resting phase of 48 h. To induce foam cell formation, after the resting phase, 50 mg/L acLDL, prepared as described below, in an RPMI 1640 medium containing 5% FBS and 1% PS was added, and cells were incubated for 48 h.

### 4.2. HDL Isolation, APOA1 Purification, and LDL Isolation and Modification

LDL (1.019 < d < 1.063 g/mL) and HDL (1.063 < d < 1.21 g/mL) were isolated from fresh human normolipidemic plasma of blood donors via sequential ultracentrifugation as previously described [46]. Lipid-free APOA1 was isolated from HDL via delipidation and ion-exchange HPLC [47]. Reconstituted HDL (rHDL) was prepared using the sodium cholate dialysis method [48], with minor modifications. The complexes were prepared at an APOA1:POPC molar ratio of 1:80. Briefly, POPC and cholesterol were mixed and dissolved in chloroform:methanol (2:1, *v*/*v*). The solvent was evaporated under nitrogen. After drying, the lipids were suspended in 10 mm Tris-HCl (pH 8.0), 0.15 m NaCl, and 1% sodium EDTA (*w*/*w)* by vortexing and incubating on ice for 1 h. Sodium cholate was added to a final cholate/POPC molar ratio of 1:1, and the mixture was incubated for 1 h on ice. The appropriate amount of apolipoprotein was then added, followed by a 1 h incubation on ice. The sodium cholate and lipid-free APOA1 were removed via extensive dialysis against a 10 mM Tris-HCl (pH 8.0), 0.15 M NaCl buffer at 4 °C using tubing with a molecular mass cut-off of 50 kDa. For LDL acetylation, 30 mg LDL was diluted in a 5 mL 0.9% NaCl buffer, and 5 mL saturated NaOAc was added. After 15 min, 70 µL acetic anhydride was added step-wise. After further incubation for 30 min, acLDL was dialyzed three times against a 0.9% NaCl buffer.

### 4.3. Cell Surface Capture of Plasma Membrane Proteins

The auto-CSC experiments with THP1 cells, activated THP1 cells, foam cells, HEPG2 cells, EA.hy926 cells, and HAECs were performed as previously described [15]. In brief, around 20 million cells were used per sample. For each condition, triplicates were processed. Cells were oxidized with 3 mM sodium periodate in PBS (pH 6.5) with 0.1% FBS for 15 min at 4 °C in the dark and labeled with 4 mM biocytin hydrazide using 5 mM 5-methoxyanthranilic acid as catalyst in PBS (pH 7.4) for 1 h at 4 °C. After every step, cells were washed with PBS. For adherent cells, cells were harvested by scraping. Pelleted cells were lysed in 500 μL 50 mM ammonium bicarbonate with 0.2% Rapidgest (Waters, Milford, CT, USA) using four 30 s sonication pulses in a VialTweeter (Dr. Hielscher, Teltow, Germany). Subsequently, samples were reduced, alkylated, and digested with trypsin from the bovine pancreas (Sigma-Aldrich, St. Louis, MO, USA). The streptavidin capture (80 µL Pierce™ Streptavidin UltraLink™ Resin, ThermoFisher Scientific, Waltham, MA, USA) of biotinylated peptides, washing, and PNGase (NEB, Ipswich, MA, USA) release was performed with a Versette liquid handling robot (ThermoFisher Scientific, Waltham, MA, USA). Sample purification was performed over C18 resin (100 µg capacity columns or plates; The Nest Group, Ipswich, MA, USA) using 2% to 80% acetonitrile with 0.1% formic acid (Chemie Brunschwig, Basel, Switzerland).

### 4.4. HATRIC-Based Identification of HDL Receptors

HATRIC-LRC was performed as described elsewhere [16]. For each condition, triplicates were processed. Ligand–HATRIC coupling was performed with 100 μg of each ligand (TRFE, APOA1, rHDL, or HDL) incubated together with 70 μg of HATRIC (100 mM stock solution in DMSO) for 2 h with slow rotation at 22 °C in 100 μL 25 mM HEPES (pH 8.2). Cells were washed with PBS (pH 6.5) and oxidized with 1.5 mM NaIO_4_ for 15 min at slow rotation at 4 °C in the dark. Subsequently, cells were washed with PBS (pH 7.4) and incubated in 10 mL PBS (pH 7.4) containing 5 mM catalyst 5-methoxyanthranilic acid (Sigma-Aldrich, St. Louis, MO, USA) and the ligand coupled to HATRIC. After incubation for 90 min at 4 °C in the dark, cells were scraped into PBS (pH 7.4) and transferred to 1.5 mL tubes.

Pelleted cells were lysed in 500 μL 8 M urea (pH 8.0) with 0.2% Rapidgest (Waters, Milford, CT, USA) and protease inhibitors (Roche, Basel, Switzerland) using three 20 s sonication pulses in a VialTweeter (Dr. Hielscher, Teltow, Germany). Per replicate, 100 μL alkyne agarose beads (Click-iT™ Protein Enrichment Kit, ThermoFisher Scientific, Waltham, MA, USA) were washed three times with 1.8 mL MilliQ water. The lysates were added to the beads, and a 500 μL ml 2× Click Chemistry Buffer (2 mM CuSO_4_, 12 mM THPTA, and 20 mM sodium ascorbate) was added. The copper-catalyzed azide-alkyne cycloaddition reaction was conducted for 18 h on a Versette liquid-handling robot (ThermoFisher Scientific, Waltham, MA, USA) at room temperature. After incubation, alkyne agarose beads were washed with 1.8 mL MilliQ water in 8 M urea, 1 M NaCl, and 100 mM Tris (pH 8). The reduction was performed with 5 mM Tris(2-carboxyethyl)phosphine for 30 min at room temperature, and alkylation was performed with 40 mM iodoacetamide for 30 min at room temperature in the dark.

Beads were then washed six times with 8 M urea and 3 M NaCl, four times with 80% isopropanol, six times with 100 mM NaHCO_3_ (pH 11) and 50 mM ammonium bicarbonate, and four times with 20% acetonitrile, water, and 50 mM ammonium bicarbonate. For tryptic digestion, beads were incubated in 400 μL 50 mM ammonium bicarbonate containing 4 μL of sequencing-grade modified trypsin (Promega, Fitchburg, MA, USAUSA) at 37 °C overnight. The tryptic peptide fraction was collected, acidified with 10% formic acid to pH 3–4, and desalted over UltraMicroSpin C18 Columns ( The Nest Group, Ipswich, MA, USA) with 5–60 μg capacity.

### 4.5. Liquid Chromatography with Tandem Mass Spectrometry (LC-MS/MS) Analyses

For mass spectrometry (MS) measurements, peptide samples were reconstituted in 3% acetonitrile, with 0.1% formic acid in HPLC-grade water, and 1 µg of the sample was loaded onto an EASY-nano-HPLC system (EASY-nLC 1000, ThermoFisher Scientific, Waltham, MA, USA) equipped with a reverse-phase column (75 μm ID) packed in-house with 15 cm stationary phase (Reprosil-Pur C18-AQ 1.9 μm, 200 Å, Dr. Maisch, Ammerbuch, Germany). The HPLC was coupled to an Orbitrap Fusion Tribrid Mass Spectrometer (ThermoFisher Scientific, Waltham, MA, USA) equipped with a nano-electrospray ion source (ThermoFisher Scientific, Waltham, MA, USA). For HATRIC-LRC, peptides were loaded onto the column with buffer A (0.1% formic acid) and eluted with a 60 min gradient of 5–28% buffer B (99.9% ACN, 0.1% formic acid), followed by a 5 min gradient from 28–45% buffer B, and two subsequent washing steps with 80% buffer B. The MS was operated in a data-dependent manner with high-resolution MS1 at 120,000 within either 375 to 1500 m/z or 395 to 1250 m/z. The maximum injection time was set to 50 ms. Precursors were excluded from fragmentation after being selected 2 or 3 times within 30 s. For MS/MS acquisition, the intensity threshold was set to 5.0 × 10^3^. Precursor ions were fragmented using collision-induced dissociation at 35% with Iontrap detection at a rapid scan rate (isolation width 1.6 m/z, normalized AGC target 20%). Cycle time was set to 3 s.

For data analysis, RAW data files were converted into mzML using MSconvert. Fragment ion spectra were searched with COMET (v27.0) against UniprotKB (Swiss-Prot, Homo sapiens from March 2019) containing common MS contaminants and standards. The precursor mass tolerance was set to 20 ppm. Search parameters were fully tryptic for LRC and semi-tryptic for CSC with carbamidomethylation as a fixed modification for cysteines. Oxidation of methionine and deamidation of asparagine were set as variable modifications. Probability scoring was performed with the Trans-Proteomic Pipeline (v4.6.2) using PeptideProphet. Peptides with an error rate of ≤1% were selected for quantification. For CSC, peptide identifications were further filtered for the presence of the consensus NXS/T sequence with the simultaneous deamidation (+0.98 Da) of asparagines. Non-conflicting peptide intensities were used in Progenesis QI (Nonlinear Dynamics, Newcastle, UK) for label-free MS1-based quantification. For CSC snapshots, peptide intensities were quantified in Progenesis and the scaled rankings of the quantified proteins per cell line were used for cross-comparison between cell lines.

### 4.6. Statistical Data Evaluation and Visualization

For statistical data evaluation, Progenesis results were processed with the SafeQuant R package [49]. The total MS1 peak-normalized and summarized peptide expression values were used for the statistical testing of differential abundance between conditions. Further, the empirical Bayes moderated *t*-tests were applied, as implemented in the R/Bioconductor limma package [50]. The resulting per protein and condition comparison *p* values were subsequently adjusted for multiple testing using the Benjamini–Hochberg method with a fold-change cut-off of 1.5.

For the Gene Ontology enrichment analysis, a hypergeometric test was performed using the R package GOFuncR [51]. The identified proteins on untreated HAECs were used as a background, and Gene Ontology enrichment was performed on proteins significantly up- and downregulated in HAECs through treatment with VEGF-A. The family-wise error rate was estimated with 1000 random sets.

Data were processed in R (v4.0.2.) and visualized with the ggplot2, gplots, UpSetR, and prcomp packages. In Figure 4A proteins were annotated as HDL or surface derived by using an HDL proteome list [1] and an in silico surfaceome resource [31]. Voronoi treemaps were created using the Foam Tree tool as previously described [31]. Illustrations in Figure 1A,B were created with BioRender.com.

### 4.7. MERTK and SCRB1 Silencing

HAECs were transfected with siRNA targeted to *MERTK*, *SCRB1*, or non-silencing control siRNA (SMARTpool, Dharmacon, Lafayette, USA) at a final concentration of 10 nmol/L using a Lipofectamine RNA IMAX transfection reagent (13778150, Invitrogen, Waltham, MA, USA) in an antibiotic-free growth medium. All experiments were performed 72 h post-transfection, and the efficiency of transfection was confirmed using quantitative RT-PCR (Appendix A) and Western blotting (Appendix A).

For *MERTK* and *SCRB1* silencing in EA.hy926 cells, lentiviral shuttle plasmids for the expression of shRNA targeting human *MERTK* (TRCN0000000865, pLKO.1), shRNA targeting *SCRB1* (TRCN0000056966, pLKO.1), or a control shRNA (a gift from David Sabatini; Addgene plasmid #1864; http://n2t.net/addgene:1864 (accessed on 14 August 2022); RRID: Addgene_1864) [52] were used. Lentivirus was packaged according to the following protocol: Shuttle plasmid (8 µg), psPAX2 packaging vector (2 µg, a gift from Didier Trono; Addgene plasmid #12260; http://n2t.net/addgene:12260 (accessed on 14 August 2022); RRID: Addgene_12260), and pMD2G envelope plasmid (4 µg, a gift from Didier Trono; Addgene plasmid #12259; http://n2t.net/addgene:12259 (accessed on 14 August 2022); RRID: Addgene_12259) were transfected into HEK-293 T cells (ThermoFisher Scientific, Waltham, MA, USA) with 1:3 DNA:polyethylenimine. After 12 h, a fresh medium was added (DMEM containing 10% FBS and 1% penicillin–streptomycin). After 48 h, the supernatant was collected, filtered through a 0.2 µm filter (Sarstedt AG, Nümbrecht, Germany), and aliquoted. The transduction into EA.hy926 cells was performed with polybrene (8 µg/mL). Cells were selected with puromycin for 3 passages before the experiments. The depletion of the proteins of interest was confirmed with RT-qPCR (Appendix A) and Western blotting (Appendix A).

### 4.8. Quantitative Real-Time PCR and Western Blot

The total RNA was isolated using an RNeasy mini kit (74104, QIAGEN, Venlo, The Netherlands) according to the manufacturer’s instructions. The genomic DNA was removed via digestion using DNase (Roche, Basel, Switzerland) in the presence of an RNase inhibitor (Ribolock, ThermoFisher Scientific, Waltham, MA, USA). Reverse transcription was performed using M-MLVRT (200 U/μL, Invitrogen, Waltham, MA, USA), following the manufacturer’s suggested protocol. Quantitative PCR was performed with Lightcycler FastStart DNA Master SYBR Green I (Roche, Basel, Switzerland) using gene-specific primers as follows: *SCRB1* (forward: CTG TGG GTG AGA TCA TGT GG; reverse: GCC AGA AGT CAA CCT TGC TC), *MERTK* (forward: CGG CGA GCC ATT GAA CTT AC; reverse: GAC CCA AAC TCT CCT TCA CCC) normalized to *GAPDH* (forward: CCC ATG TTC GTC ATG GGT GT; reverse: TGG TCA TGA GTC CTT CCA CGA TA). For Western blotting, a RIPA buffer was used to prepare protein lysates. Proteins were separated on a 4–12% NuPAGE Bis-Tris (Invitrogen, Waltham, MA, USA). After blotting, antibodies to MERTK (D21F11, Cell Signaling, Danvers, USA), SCRB1 (nb400_104, Novus, Centennial, CO, USA), and anti-GAPDH (clone GAPDH-71.1, Sigma-Aldrich, St. Louis, MO, USA) were applied for protein detection. To determine the state of MERTK-phosphorylation upon HDL binding, antibodies to MERTK (Cell Signaling, D21F11), phospho-MERTK (Fabgennix, FGX-PMKT-140AP), and anti-tubulin (T5168, Sigma-Aldrich, St. Louis, MO, USA) were used (Appendix A).

### 4.9. HDL Binding and Association Experiments

The quantification of cellular binding and of association (i.e., uptake) of radiolabeled HDL into endothelial cells was performed as previously described [53]. Experiments were performed in DMEM (Sigma-Aldrich, St. Louis, MO, USA) containing 25 mmol/L HEPES and 0.2% BSA. Cells were incubated with 10 μg/mL of ^125^I-HDL without or with a 40-fold excess of unlabeled HDL (unspecific) for 1 h at 4 °C for cellular binding and for 1 h at 37 °C for association experiments. Specific cellular binding or association was calculated by subtracting the values obtained in the presence of excess unlabeled HDL (unspecific) from those obtained in the absence of unlabeled HDL (total).

### 4.10. Proximity Ligation Assay

Duolink™ proximity ligation assays were performed with a Duolink In Situ Red Starter Kit Mouse/Goat (Sigma-Aldrich, St. Louis, MO, USA) according to the manufacturer’s protocol. Cells were seeded at a density of 80–90% into chamber slides (Nunc) and fixed after 24 h with 4% paraformaldehyde at room temperature for 15 min. After sample b, locking, the primary antibodies mouse-anti-human SCARB1 (1:100; clone m1B9, Biolegend, San Diego, CA, USA) and goat-anti-human MERTK (1:40; AF891, Novus, Centennial, CO, USA) were applied overnight at 4 °C. Control slides were incubated either with a buffer without primary antibodies or with a single anti-SCRB1 or anti-MERTK antibody only. After washing, the two secondary antibody PLA probes (diluted 1:5) were applied for 1 h at 37 °C. Ligation was performed with 1:40 ligase at 37 °C for 30 min. Polymerase was applied at 1:80 to the samples for 100 min at 37 °C for signal amplification. After washing, the samples were mounted with a mounting medium containing DAPI. Slides were imaged on a Leica SPE confocal laser scanning microscope. The red fluorescent signal was excited at 532 nm, and the emission filter was set to 583–620 nm bandwidth. DAPI settings were 405 nm for excitation and 430–480 nm for emission. Confocal slices of 116.4 × 116.4 × 0.5 µm^3^ were recorded with a 63× oil objective (NA 1.3) and a pinhole at 1 airy unit for both channels.

### 4.11. Nile Red Staining

Cells were stained with Nile red (5 µg/mL, ThermoFisher Scientific, Waltham, MA, USA) at 4 °C for 15 min, washed three times with PBS, and then fixed with 4% paraformaldehyde for 10 min. Coverslips were mounted with ProLong™ Diamond Antifade Mountant with DAPI (Invitrogen, Waltham, MA, USA). Cells were imaged on a Leica SPE confocal laser scanning microscope using a 40× oil objective (NA 1.15). DAPI excitation/emission was at 405 nm/430–480 nm, and that of Nile red was at 532 nm/549–586 nm. Confocal frames had a dimension of 183.3 × 183.3 µm^2^.

## Figures and Tables

**Figure 1 ijms-23-09506-f001:**
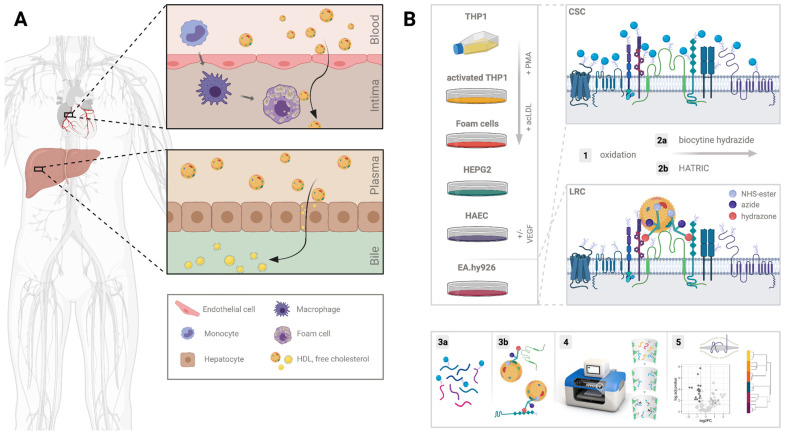
Schematic illustration of the rationale for the selection of the cell types and the technologies used for characterization of the receptor interface of HDL: (**A**) in reverse cholesterol transport, HDL crosses the endothelial barrier and takes up free cholesterol from lipid-laden macrophages in the intima of blood vessels. HDL then travels via the lymph to the liver to deliver cholesterol for biliary excretion; (**B**) THP1 monocytes, PMA-activated THP1 macrophages, acLDL-treated THP1 foam cells, HEPG2 hepatocytes, human aortic endothelial HAECs, and EA.hy926 microvascular endothelial cells were used as cellular model systems to investigate the potential HDL-interacting cellular surfaceome. (**1**) Cells were first mildly oxidized to (**2a**) biotinylate the surfaceome for auto-CSC or (**2b**) to tag receptors proximal to HDL using HATRIC-LRC. (**3a**,**3b**) For auto-CSC, proteins were digested and biotinylated peptides were enriched and (**3b**) for HATRIC-LRC proteins were on-bead digested (**4**) on an automated liquid handling system. (**5**) Peptides were identified and quantified via mass spectrometry.

**Figure 2 ijms-23-09506-f002:**
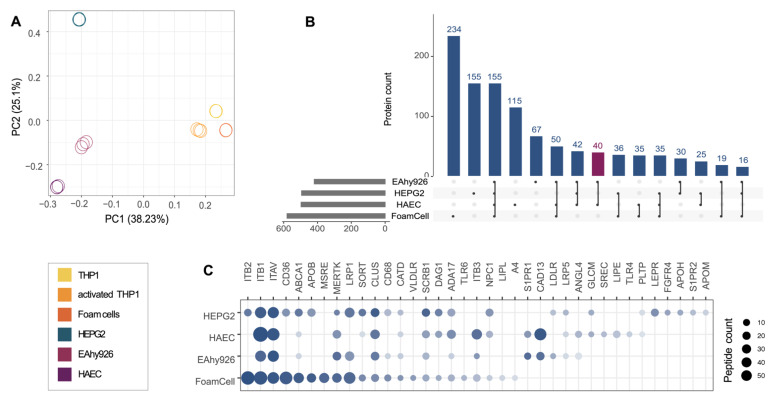
Surfaceome characterization of RCT-relevant cellular model systems: (**A**) principal component analysis of protein abundance levels from six different cell types; (**B**) upset plot illustrates the overlap of the identified proteins per RCT-relevant cell type. The purple bar highlights the specific endothelial surfaceome; (**C**) proteins with a Gene Ontology (biological processes) term associated with processes relevant for cholesterol or lipoprotein metabolism. MERTK, identified as an HDL co-receptor using HATRIC-LRC, is displayed as well. Dot size corresponds to the number of quantified peptides for the different cell populations and color grade to the ranked and scaled abundance of the proteins in the respective experiments.

**Figure 3 ijms-23-09506-f003:**
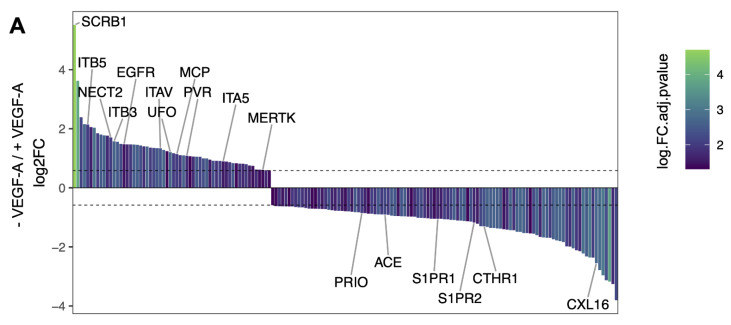
VEGF-A triggers surfaceome remodeling of HAECs: (**A**) waterfall plot of all significantly affected proteins on HAECs in cells treated with VEGF-A versus untreated cells. *Y*-axis represents log2 fold change (FC), and bars are colored according to FC adjusted *p* value. Only significantly regulated proteins are depicted (FC > 1.5, FC adjusted *p* value < 0.05) (see Appendix A for details). Labeled proteins belong to the top GO term of each protein group (for upregulated proteins: virus receptor activity; for downregulated proteins: G protein-coupled receptor binding) as well as MERTK, which was associated with apoptotic cell clearance, regulation of phagocytosis, and regulation of vesicle-mediated transport; (**B**) GO analysis of proteins in HAECs up- and downregulated through VEGF-A treatment. The top 15 GO terms (ranking based on family-wise error rate (FWER)) are named in the graph. Non-significant terms with a *p* value > 0.01 and cellular component terms were excluded from the plot. The size of the dot corresponds to the log.FWER (i.e., smaller FWER corresponds to larger dots).

**Figure 4 ijms-23-09506-f004:**
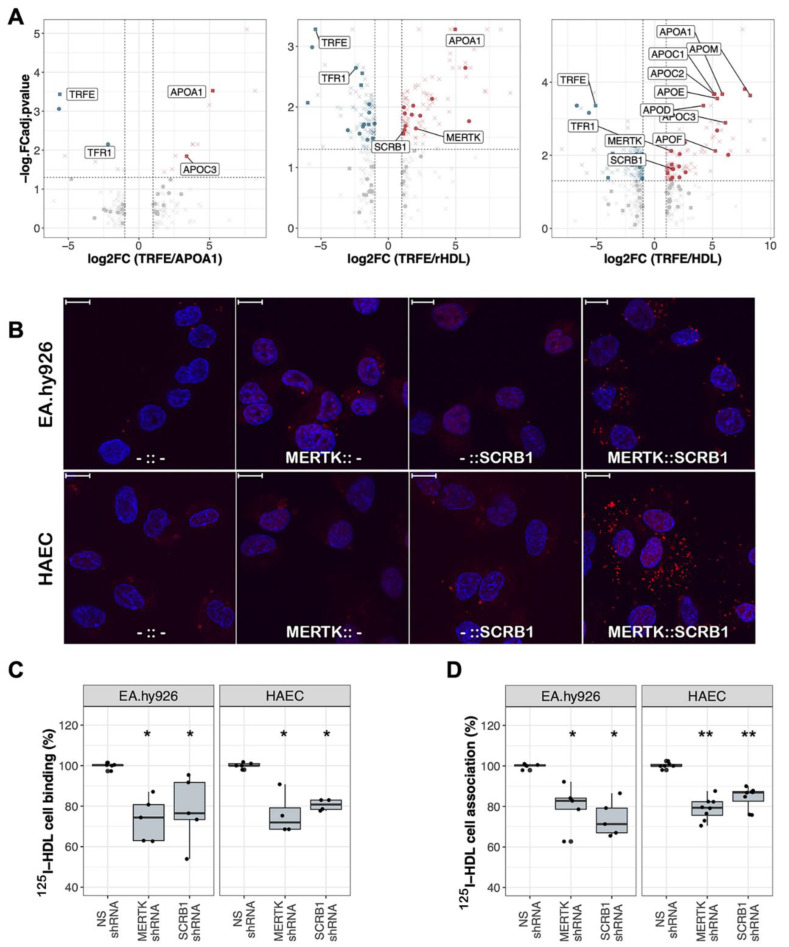
MERTK is a novel receptor of HDL: (**A**) volcano plots showing HATRIC-LRC results for EA.hy926 cells treated with the ligands APOA1, rHDL, or native HDL quantified relative to the TRFE-treated negative control. Significant protein changes are colored in red (upregulated) or blue (downregulated). Proteins annotated as HDL derived are displayed as points and proteins annotated as surface derived are displayed as squares. All other proteins are displayed as crosses. FC was set > 2, FC adjusted *p* value < 0.05; (**B**) confocal slices illustrating the outcomes of the proximity ligation assays (PLAs) with anti-SCRB1 and anti-MERTK antibodies on EA.hy926 cells and HAECs (scale bars 10 μm). From left to right: proximity ligation reaction with no primary antibody (-::-), anti-MERTK antibody only (MERTK::-), anti-SCRB1 antibody only (-::SCRB1) (all controls), and both primary antibodies (MERTK::SCRB1). Red: PLA signal. Blue: DAPI; (**C**,**D**) percent HDL (**C**) binding and (**D**) association (binding plus uptake) after silencing of *MERTK* or *SCRB1* compared with the non-silencing (NS) control cells treated with non-targeted shRNA or siRNA. The data are presented as boxplots showing the minimum and maximum values (vertical line), first and third quartile (box), and median (horizontal line). Each boxplot includes 4–8 independent experiments with four replicates each. The significance of MERTK or SCRB1 depletion compared with the control was assessed with the Wilcoxon rank-sum test. (* *p* < 0.05, ** *p* < 0.01).

## Data Availability

Mass spectrometry raw files and processed data were deposited into the ProteomeXchange Consortium database (PXD022291) via the MassIVE partner repository (https://massive.ucsd.edu) with the dataset identifier MSV000086397.

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
