# Peer review of "Decoding Functional High-Density Lipoprotein Particle Surfaceome Interactions"

_ijms, 2022, doi:10.3390/ijms23169506_

Round 1

Reviewer 1 Report

Figure 2B: Heat map as presented has no meaning. Please indicate the clustering of genes.

Figure 3B: Blue circles: some are free while others are attached to a square with a line. Describe the meaning of this presentation. 

Figure 4B; First set of pics need description of the symbol shown in the pic.

Reviewer 2 Report

The authors describe their work on the characterization and understanding of the functional interplay between HDL particles and tissue. The data provided direct evidence for a complex and dynamic HDL receptome and that receptor nanoscale organization may influence functional HDL binding and uptake. This is an interesting study. Appropriate methodology has been employed and the conclusions appear to be justified based on the data at hand. I have a few recommendations for consideration.

1. Introduction. Please provide a clear hypothesis to be tested in the study.

2. Results. Fig 4. Please indicate areas of interest with arrows.

3. Discussion. The authors should elaborate and emphasize the novelty aspect of their work.

4. Discussion. With the complexity of the HDL receptome in mind, can the authors describe in more detail the clinical applicability of their findings. Hoe could this be tested in vivo?

5. General. Please review formatting of the reference section and that it is in compliance with journal requirements. 
